# Serum Metabolomic Alteration in Rats with Osteoarthritis Treated with Palm Tocotrienol-Rich Fraction Alone or in Combination with Glucosamine Sulphate

**DOI:** 10.3390/life13122343

**Published:** 2023-12-15

**Authors:** Sophia Ogechi Ekeuku, Jen-Kit Tan, Hiba Murtadha Al-Saadi, Fairus Ahmad, Mohd Ramli Elvy Suhana, Azlan Mohd Arlamsyah, Fadhlullah Zuhair Japar Sidik, Juliana Abdul Hamid, Soelaiman Ima-Nirwana, Kok-Yong Chin

**Affiliations:** 1Department of Biochemistry, Faculty of Medicine, Universiti Kebangsaan Malaysia, Cheras 56000, Malaysia; sogechie@ukm.edu.my (S.O.E.);; 2Department of Pharmacology, Faculty of Medicine, Universiti Kebangsaan Malaysia, Cheras 56000, Malaysia; 3Department of Anatomy, Faculty of Medicine, Universiti Kebangsaan Malaysia, Cheras 56000, Malaysia

**Keywords:** amino acid metabolism, arthritis, joint, tocotrienol, vitamin E

## Abstract

Osteoarthritis (OA) is a degenerative joint condition with limited disease-modifying treatments currently. Palm tocotrienol-rich fraction (TRF) has been previously shown to be effective against OA, but its mechanism of action remains elusive. This study aims to compare serum metabolomic alteration in Sprague–Dawley rats with monosodium iodoacetate (MIA)-induced OA which were treated with palm TRF, glucosamine sulphate, or a combination of both. This study was performed on thirty adult male rats, which were divided into normal control (*n* = 6) and OA groups (*n* = 24). The OA group received intra-articular injections of MIA and daily oral treatments of refined olive oil (vehicle, *n* = 6), palm TRF (100 mg/kg, *n* = 6), glucosamine sulphate (250 mg/kg, *n* = 6), or a combination of TRF and glucosamine (*n* = 6) for four weeks. Serum was collected at the study’s conclusion for metabolomic analysis. The findings revealed that MIA-induced OA influences amino acid metabolism, leading to changes in metabolites associated with the biosynthesis of phenylalanine, tyrosine and tryptophan as well as alterations in the metabolism of phenylalanine, tryptophan, arginine and proline. Supplementation with glucosamine sulphate, TRF, or both effectively reversed these metabolic changes induced by OA. The amelioration of metabolic effects induced by OA is linked to the therapeutic effects of TRF and glucosamine. However, it remains unclear whether these effects are direct or indirect in nature.

## 1. Introduction

Osteoarthritis (OA) is a degenerative joint disease characterised by cartilage degradation, synovitis, subchondral bone alternations and osteophyte formation [1]. Joint oedema, tenderness, and pain are frequently present in patients with OA, and these conditions limit their joint motion [2,3]. According to the Global Burden of Disease Study, 528 million people worldwide suffer from OA, which represented a 27.5% increase since 2010. Although OA does not directly cause death, it was the 15th major cause of years spent living with a disability worldwide in 2019 [4]. Patients with late-stage OA will require an expensive and invasive total arthroplasty with potential side effects such as infection and bleeding if it is not managed properly [5].

Traditional pharmacological agents for OA, such as intra-articular corticosteroid and hyaluronan injections, as well as oral analgesics, focus on symptomatic relief [6,7]. Glucosamine sulphate, a substrate for the synthesis of proteoglycans, is a commonly used alternative treatment for OA [8]. Numerous meta-analyses have found that glucosamine sulphate may reduce OA pain while having only minor effects on patients’ knee functions [9,10]. Furthermore, glucosamine sulphate exerts potent anti-inflammatory actions by inhibiting nuclear factor kappa B translocation and promoting cyclooxygenase enzyme degradation [11]. The search for better joint protecting agents is still ongoing.

Tocotrienol is one of the natural compounds that has received attention for its effects against OA recently. It is a member of the vitamin E (tocochromanols) family and has a structure similar to tocopherol, consisting of a six-carbon chromanol ring and a hydrophobic carbon side chain. The isomers of tocotrienol (α, β, γ, and δ) are distinguished based on the position of the side chains on the chromanol ring [12,13]. Tocotrienol, in varying compositions of its isomers, can be found abundantly in palm oil, annatto beans, and rice bran [12]. Tocotrienol and its metabolites have various biological activities, most notably anti-inflammatory effects [14]. Previous studies found that γ-tocotrienol [15], δ-tocotrienol [16] and palm tocotrienol-rich fraction (TRF) [17] protected against inflammatory and rheumatoid arthritis in rats. A study in rats with monosodium iodoacetate (MIA)-induced OA has reported the effectiveness of annatto tocotrienol in preventing cartilage and subchondral bone changes [18]. Palm TRF was also reported to reduce the serum level of cartilage oligomeric matrix protein and increase grip strength when combined with glucosamine in OA rats [19]. An open-label study supplemented palm TRF in patients with OA and found some positive outcomes in pain and functional scores, but the joint structural outcomes were not tested [20]. Despite these studies, the anti-OA mechanism of tocotrienol remains elusive.

Metabolites are currently studied as biomarkers of OA diagnosis, prognosis and therapeutics. Metabolites in the blood and synovial fluid may serve as biomarkers for OA [21]. The development of metabolomics offers new ways to comprehend the nature of disease and monitor treatment [22]. Therefore, this study aims to compare metabolomic changes in rats with OA induced by MIA and treated with palm TRF, glucosamine sulphate, or both agents combined.

## 2. Results

The present study included five experimental groups: namely, the sham control group (SC) and OA control group (OC) treated with refined olive oil, the OA group treated with palm TRF (OT), glucosamine sulphate (OG), or a combination of palm TRF and glucosamine sulphate (OGT). 

### 2.1. Differential Metabolomic Analysis among the Study Groups

To perform a metabolomic analysis comparing the control and treated groups, principal component analysis (PCA) was first conducted to visually assess the presence of any discernible separation among the various groups. Its purpose is to highlight variations and reveal prominent patterns within a dataset [23]. Each point symbolises all the metabolites expressed in a specific sample from an individual rat in each treatment group, facilitating comparisons between groups as similar expression profiles cluster together. Principal component 1 (PC1) captures the highest variation in the data, while PC2 represents the second-highest variation. This facilitates a clearer visualisation of the similarity in metabolite variation between groups.

The PCA score plot generated for all groups in the negative mode showed a 27.4% variation, whereby the first principal component (PC1) score was 17.1% and the second principal component (PC2) score was 10.3% (Figure 1A). The PCA score plot generated for all groups in the positive mode similarly showed 26.2% variation, with a PC1 value of 15% and PC2 value of 11.2% (Figure 1B). There was no clear separation when comparing all the groups together, suggesting the complex and similar metabolite profiles among these groups. For better visualisation, two-group comparisons were made. The PCA score plot generated for osteoarthritic control (OC) and sham control (SC) showed a 41.4% variation where the PC1 score was 24.2% and PC2 score was 17.2% in the negative mode (Figure 1C), while the positive mode showed a 34.3% variation with PC1 value of 17.9% and PC2 value of 16.4% (Figure 1D). There was a distinct separation in the positive and negative modes between OC and SC. This indicates that as an established osteoarthritic model, the metabolite expression patterns of OC differ from those of SC.

Then, the distributions of samples based on their metabolomic profiles for treated versus vehicle-treated groups were assessed. The PCA score plot generated for the glucosamine sulphate-treated OA group (OG) and OC showed a 40.8% variation where the PC1 score was 24.6% and the PC2 score was 16.2% in the negative mode (Figure 2A), while the positive mode showed a 35.1% variation with a PC1 value of 19.9% and a PC2 value of 15.2% (Figure 2B). The PCA score plot generated for the palm TRF-treated OA group (OT) and OC showed a 38.3% variation where the PC1 score was 20.2% and the PC2 score was 18.1% in the negative mode (Figure 2C), while the positive mode showed a 35.5% variation with a PC1 value of 20.9% and a PC2 value of 14.6% (Figure 2D). The PCA score plot generated for the OA group treated with glucosamine sulphate and palm TRF (OGT) and OC showed a 40.8% variation where the PC1 score was 21.3% and the PC2 score was 19.5% in the negative mode (Figure 2E), while the positive mode showed a 33.9% variation with a PC1 value of 19.3% and a PC2 value of 14.6% (Figure 2F). Partial separation between OG vs. OC and OGT vs. OC was noted in both positive and negative modes, indicating a subtle resemblance in metabolite expression patterns. Conversely, no discernible separation was observed in the comparisons of OT vs. OC in both positive and negative modes, suggesting similarities in their metabolite expression patterns.

Next, the variations among the different treatment groups were inspected. The PCA score plot generated for OT and OG showed a 39.6% variation where the PC1 score was 23.1% and the PC2 score was 16.5% in the negative mode (Figure 3A), while the positive mode showed a 37.9% variation with a PC1 value of 23% and a PC2 value of 14.9% (Figure 3B). The PCA score plot generated for OGT and OG showed 39.6% variation where the PC1 score was 26.4% and the PC2 score was 13.2% in the negative mode (Figure 3C), while the positive mode showed 35% variation with a PC1 value of 22% and a PC2 value of 13% (Figure 3D). The PCA score plot generated for OGT and OT showed a 38.3% variation where the PC1 score was 21% and the PC2 score was 17.3% in the negative mode (Figure 3E), while the positive mode showed a 35.9% variation with a PC1 value of 19.2% and a PC2 value of 16.7% (Figure 3F). Partial separation between OT vs. OG was noted in both positive and negative modes, indicating a subtle resemblance in metabolite expression patterns. Conversely, no discernible separation was observed in the comparisons of OGT vs. OG and OGT vs. OT in both positive and negative modes, suggesting similarities in their metabolite expression patterns. 

### 2.2. Metabolite Profiling and Regulation in Treated and Control Groups

To study the effect of OA, the metabolomic profile of the OC group was compared to that of the SC group. The comparison between OC and SC identified 44 differentially expressed metabolites from 74 metabolic features. The fold-change comparison showed that 2,2′-methylenebis(4-methyl-6-tert-butylphenol) was the most downregulated metabolite with a 10.19-fold decrease, which was followed by 3-hydroxybutyric acid with a 2.31-fold decrease and dehydronorketamine with a 1.35-fold decrease in the OC group. The most upregulated metabolite in the OC group was 2-hydroxyhippuric acid with a 124.02-fold increase, which was followed by 5-sulfosalicylic acid with a 79.74-fold increase and azelaic acid with a 16.14-fold increase (Table 1). 

To study the effect of treatments on OA, the metabolomic profile of the treatment group was compared to that of the OC group. The comparison between OG and OC identified 35 differentially expressed metabolites from 74 metabolic features. Of note, 2-hydroxyhippuric acid was the most downregulated in the OG group with an 86.20-fold decrease, which was followed by 5-sulfosalicylic acid with a 38.92-fold decrease and xylazine with a 6.51-fold decrease. The most upregulated metabolite in the OG group was 2,2′-methylenebis(4-methyl-6-tert-butylphenol) with a 5.41-fold increase, which was followed by 3,4-dihydroxybenzenesulfonic acid with a 2.21-fold increase and 3-hydroxybutyric acid with a 1.44-fold increase (Table 1). The comparison between OT and OC found 23 differentially expressed metabolites from 74 metabolic features. Azelaic acid was the most downregulated in the OT group with a 4.48-fold decrease, which was followed by (15Z)-9,12,13-trihydroxy-15-octadecenoic acid with a 3.22-fold decrease and phenazone with a 3.18-fold decrease. The most upregulated metabolite in the OT group was 2,2′-methylenebis(4-methyl-6-tert-butylphenol) with a 5.65-fold increase, which was followed by 4-phenylbutyric acid with a 1.99-fold increase and 3-hydroxybutyric acid with a 1.52-fold increase (Table 1). The comparison between OGT and OC yielded 28 differentially expressed metabolites from 74 metabolic features. Azelaic acid was the most downregulated in the OGT group with a 14.12-fold decrease, which was followed by (15Z)-9,12,13-trihydroxy-15-octadecenoic acid with a 6.34-fold decrease and 2-hydroxyhippuric acid with a 5.82-fold decrease. The most upregulated metabolite in the OGT group was 4-phenylbutyric acid with a 2.78-fold increase, which was followed by pantothenic acid with a 1.10-fold increase and tyrosine with a 1.07-fold increase (Table 1). 

To study the differences in metabolic profile between the three treatment regimens, OG, OT and OGT were compared to each other. The comparison between OT and OG identified 20 differentially expressed metabolites from 74 metabolic features. Of note, 3,4-dihydroxybenzenesulfonic acid was the most downregulated with a 1.82-fold decrease, which was followed by 4′-(Imidazol-1-yl)acetophenone with a 1.54-fold decrease and pipecolic acid/pipecolinic acid/nipecotic acid with a 0.48-fold decrease. The most upregulated metabolite was 2-hydroxyhippuric acid with a 78.84-fold increase, which was followed by 5-sulfosalicylic acid with a 48.56-fold increase and xylazine with a 6.42-fold increase (Table 1). A comparison between OGT and OG yielded 17 differentially expressed metabolites from 74 metabolic features. There were no significantly downregulated metabolites. The most upregulated metabolite was 4-phenylbutyric acid with a 2.97-fold increase, which was followed by 3-indoxyl sulphate with a 2.07-fold increase and erucamide with a 1.45-fold increase (Table 1). The comparison between OGT and OG found 10 differentially expressed metabolites from 74 metabolic features. Suberic acid was the most downregulated with a 1.74-fold decrease, which was followed by N6,N6,N6-trimethyl-L-lysine with a 0.19-fold decrease. The most upregulated metabolite was erucamide with a 1.72-fold increase, which was followed by 3-indoxyl sulphate with a 1.39-fold increase and tyrosine with a 1.27-fold increase (Table 1).

**Table 1 life-13-02343-t001:** List of significantly altered metabolites in OC vs. SC, OG vs. OC, OT vs. OC, OGT vs. OC, OT vs. OG, OGT vs. OG and OGT vs. OT.

Ion Mode	Name	MW	RT	ID	OC vs. SC	OG vs. OC	OT vs. OC	OGT vs. OC	OT vs. OG	OGT vs. OG	OGT vs. OT
−	(15Z)-9,12,13-Trihydroxy-15-octadecenoic acid	330.24075	6.943	HMDB0038555	10.51	−4.68	−3.22	−6.34			
+	2-Hydroxycinnamic acid/4-Coumaric acid	164.04704	0.876	HMDB0002641	0.24	−0.25					
−	2-Hydroxyhippuric acid	195.05254	1.856	HMDB0000840	124.02	−86.20		−5.82	78.84		
+	2,2,6,6-Tetramethyl-4-piperidinol	157.14628	0.985	HMDB0031179	4.54				4.47		
−	2,2′-Methylenebis(4-methyl-6-tert-butylphenol)	340.24005	10.848	HMDB0244434	−10.19	5.41	5.65				
−	3-Hydroxybutyric acid	104.04636	0.977	HMDB0000357	−2.31	1.44	1.52				
−	3-Hydroxybutyric acid/4-Hydroxybutyric acid (GHB)	104.04634	1.037	HMDB0000357/HMDB0000710			−1.30				1.04
−	3-Indoxyl sulphate	213.00908	2.453	HMDB0000682	1.79					2.07	1.39
−	3,4-Dihydroxybenzenesulfonic acid	189.99297	1.588		2.16	2.21			−1.82		
+	**4-Hydroxy xylazine**	236.09778	3.559		4.77	−4.85			4.60		
−	4-Oxoproline	129.0416	0.917	METPA0228			−1.05	−1.72			
+	4-Phenylbutyric acid	164.08347	6.953	HMDB0000543			1.99	2.78	2.18	2.97	
+	4′-(Imidazol-1-yl)acetophenone	186.07921	3.369		1.36		−1.46		−1.54		
+	5-Methylcytosine	125.05874	0.824	HMDB0002894				−0.40			
−	5-Sulfosalicylic acid	217.98805	1.925	HMDB0011725	79.74	−38.92			48.56		
+	5′-S-Methyl-5′-thioadenosine	297.08881	3.298	HMDB0001173				0.91		0.84	1.04
+	Acetophenone	120.05742	0.875	HMDB0033910	0.35		−0.39				
+	Acetyl-L-carnitine	203.11533	1.048	HMDB0000201				−0.37			
+	Acetylcholine	145.10994	0.972	HMDB0000895		−0.27		−0.26			
+	Arginine	174.11145	0.679	HMDB0000517	0.36						
−	Azelaic acid	188.10418	5.13	HMDB0000784	16.14	−3.35	−4.48	−14.12			
+	Carnitine	161.10479	0.809	HMDB0000062	0.24	−0.35	−0.20	−0.17	0.15	0.18	
−	Cholic acid	408.2875	7.507	HMDB0000619	3.37						
+	Choline	103.09978	0.885	HMDB0000097	0.38						
+	Corticosterone	346.2134	6.976	HMDB0001547	−0.58						
+	Creatine	131.06924	0.732	HMDB0000064			−0.66		−0.47		0.62
+	Cytidine	243.08486	0.725	HMDB0000089	0.50						
+	D-Erythro-sphingosine 1-phosphate	379.24746	9.91	HMDB0000277		−0.52			0.63		
+	Dehydronorketamine	221.06048	3.791	HMDB0060549	−1.35	0.97		0.83			
−	Deoxycholic acid	392.29243	8.8	HMDB0000626	2.41						
+	Docosahexaenoic acid ethyl ester	356.27055	8.385	HMDB0251557			−1.47	−1.42			
+	Equol	242.09402	5.405	HMDB0002209					1.12		
+	Ergothioneine	229.08797	0.873	HMDB0003045	0.81	−0.76			0.87		
+	Erucamide	337.33336	13.855	HMDB0244507						1.45	1.72
−	Glutamine	146.06821	0.758	HMDB0000641		−1.22			1.27	1.40	
+	Glycerophospho-N-palmitoyl ethanolamine	453.28417	11.406		0.54	−0.50		−0.41			
+	Hexanoylcarnitine	259.17776	4.631	HMDB0000705	−0.59						
+	Hippuric acid	179.05783	3.029	HMDB0000714	1.14						
−	Histidine	155.06862	0.752	HMDB0000177						1.28	
+	Indole-3-acrylic acid	187.06296	2.294	HMDB0000734		−0.49			0.60	0.49	
+	Isoquinoline/Quinoline	129.05773	5.602	HMDB0034244/HMDB0033731						0.98	
+	**Ketamine/Esketamine**	237.09152	4.255			−0.85					
+	Kynurenine	208.08438	1.255	HMDB0000684		−0.77		−1.08			
+	L-Glutathione oxidized	612.14998	0.823	HMDB0003337	2.42	−0.51	−1.03	−0.82			
+	leucine/isoleucine/Norleucine	131.09437	0.933	HMDB0000172/HMDB0001645/HMDB0000687	0.32						
−	Malic acid	134.02056	0.783	HMDB0000744	3.55	−1.86	−1.69	−1.69			
+	Methionine	149.05076	0.802	HMDB0000696		−0.28					
+	Methyl indole-3-acetate	189.07873	5.6	HMDB0029738				0.55		1.06	
+	N6,N6,N6-Trimethyl-L-lysine	188.15208	0.948	HMDB0001325	0.65	−0.65	−0.49	−0.68	0.16		−0.19
+	Palmitoyl sphingomyelin	702.56584	21.758	HMDB0010169				0.86			
+	Palmitoylcarnitine	399.33373	10.714	HMDB0000222	0.57	−0.45		−0.40			
−	Pantothenic acid	219.11025	1.84	HMDB0000210				1.10			
+	Phenazone	188.09469	1.334	HMDB0015503	3.11		−3.18	−3.11			
−	Phenylalanine	165.07815	1.448	HMDB0000159	1.22						
+	Phenylalanine	165.07862	1.275	HMDB0000159	0.33			−0.28			
+	Pipecolic acid/Pipecolinic acid/Nipecotic acid	129.07876	0.891	HMDB0000070/HMDB0255618	−0.38				−0.48		0.30
+	Proline	115.06327	0.783	HMDB0000162	0.26	−0.37	−0.22	−0.25			
+	Propionylcarnitine	217.13103	1.436	HMDB0000824	0.67	−0.60	−0.61				
+	Pyroglutamic acid	129.04229	0.689	HMDB0000267					0.40	0.33	
+	Spermidine	145.15756	0.518	HMDB0001257	0.76	−0.51	−0.95				
+	Sphingosine	299.28154	9.435	HMDB0000252		−1.64	−0.94			1.21	
+	Stearamide	283.28667	12.646	HMDB0034146	1.89						
−	Suberic acid	174.08846	4.377	HMDB0000893	3.53	−2.08	−2.07	−3.60			−1.74
−	Taurine	125.01366	0.722	HMDB0000251	1.43						
−	Taurochenodeoxycholic acid	499.2968	7.117	HMDB0000951		−5.01					
−	Threonic acid	136.03636	0.765	HMDB0000943		1.21			1.14	1.14	
−	Threonine	119.05725	0.759	HMDB0000167	1.39						
+	trans,trans-2,4-Heptadienal	110.07319	4.068	HMDB0303844	1.43					0.82	
+	Triisopropanolamine	191.15184	0.98	mzc2688				0.59		0.65	0.43
−	Tryptophan	204.08935	2.435	HMDB0000929		−1.20	1.08	1.04			
+	Tryptophan	204.08951	2.273	HMDB0000929		−0.62			0.71	0.52	
−	Tyrosine	181.07314	0.973	HMDB0000158		−1.11		1.07		1.19	1.27
−	Uric acid	168.02756	0.9	HMDB0000289	1.42		−1.34				
+	**Xylazine**	220.10285	4.462		6.30	−6.51			6.42		

The metabolites in bold are metabolites that were introduced into the rat system through anaesthesia administration during the euthanisation process. Acronym: FC, fold change; MW, molecular weight; RT, retention time; +, increase; −, decrease; OC, osteoarthritis control; SC, sham control; OG, osteoarthritis + glucosamine; OT, osteoarthritis + tocotrienol-rich fraction; OGT, osteoarthritis + glucosamine + tocotrienol-rich fraction.

### 2.3. Pathway Analysis

Pathway analysis was performed to investigate the metabolic pathways affected by OA and the subsequent treatments. 

The effect of OA on the metabolic pathways was determined by comparing the serum metabolomic expression between SC and OC. The comparison identified 20 altered biochemical pathways. The significant pathways involved were phenylalanine, arginine and proline metabolism and phenylalanine as well as tyrosine and tryptophan biosynthesis (Figure 4; Appendix A).

The metabolic effects of treatments on rats with OA were determined by comparing the serum metabolomic expression between treatment groups and OC as well as between the treatment groups. The comparison between OG and OC found 24 altered biochemical pathways. The significant pathway involved was phenylalanine, tyrosine and tryptophan biosynthesis (Figure 5A; Appendix A). The comparison between OT and OC found 11 altered biochemical pathways. The significant pathway involved was arginine and proline metabolism (Figure 5B; Appendix A). A comparison of profiled metabolites between OGT and OC identified 14 altered biochemical pathways. The significant pathways involved were phenylalanine, tyrosine and tryptophan biosynthesis, phenylalanine metabolism and tryptophan metabolism (Figure 5C; Appendix A).

For comparison between treatments, the comparison between OT and OG identified 13 altered biochemical pathways. However, none of these pathways were significant (Figure 6A; Appendix A). The comparison between OGT and OG found 17 altered biochemical pathways. The significant pathway involved was phenylalanine, tyrosine and tryptophan biosynthesis (Figure 6B; Appendix A). The comparison of profiled metabolites between OGT and OT yielded 11 altered biochemical pathways. The significant pathway involved was phenylalanine, tyrosine and tryptophan biosynthesis (Figure 6C; Appendix A).

## 3. Discussion

In this study, the impact of OA and treatments with glucosamine and palm TRF on rats with MIA-induced OA was investigated via an untargeted metabolomic approach. The findings revealed the presence of nine distinct metabolites in the serum metabolic profile affected by OA and treatments. Subsequent analysis identified that these metabolites were associated with four specific metabolic pathways, i.e., phenylalanine, tyrosine and tryptophan biosynthesis, as well as phenylalanine, tryptophan, arginine and proline metabolism. 

Phenylalanine metabolism, as well as phenylalanine, tyrosine, and tryptophan biosynthesis pathways are integral components of amino acid metabolism. Notable metabolites affected in these pathways include L-phenylalanine, hippuric acid, and L-tyrosine. L-phenylalanine is an essential amino acid vital for normal organism development and maintenance. Previous studies have shown an upregulation of L-phenylalanine in the serum and synovial tissue of collagen-induced arthritic rats [24]. Similarly, plasma and synovial fluid phenylalanine levels have been linked to the progression of radiographic knee OA [25,26]. In the current study, L-phenylalanine was regulated in the OA rats compared to the SC, confirming the accelerated progression of knee OA. As outlined by Geisler, the phenylalanine-to-tyrosine ratio rises in the presence of inflammation, indicating a potential increase in phenylalanine levels during inflammatory processes. Given that synovial inflammation is a prominent feature of OA, it is plausible that the heightened inflammation in MIA-induced OA may have led to increased phenylalanine expression in the OC group.

Interestingly, supplementation with a combination of glucosamine and TRF led to a reduction in serum L-phenylalanine levels in the OA rats, as evidenced by its downregulation in OGT vs. OC. As per Geisler (2013), mitigating abnormalities in phenylalanine metabolism can be achieved to some extent through inflammation reduction. Glucosamine and TRF are known for their anti-inflammatory properties. Hence, it is plausible that these substances may have diminished phenylalanine expression by mitigating inflammation in OA rats. These results suggest that L-phenylalanine concentration may serve as a crucial biomarker in detecting OA and that the downregulation of phenylalanine through glucosamine + TRF supplementation could potentially improve OA. 

Hippuric acid, on the other hand, is a co-metabolite derived from phenylalanine and dietary polyphenols in mammals and microbes. Lower standardised levels of hippuric acid have been statistically linked to an increased odds ratio of being diagnosed with OA and were observed to be decreased in faecal samples of patients with OA [27]. However, increased hippuric acid levels were reported in the urine of OA progressors compared to non-progressors [28]. This aligns with the present study, where hippuric acid levels were upregulated in the serum of OC compared to SC. As suggested by Loeser et al., the difference in hippuric acid levels could be attributed to differences in biospecimen types or may indicate higher hippuric acid absorption in OA [28]. Additionally, dietary variations, particularly the consumption of foods rich in polyphenols, could also contribute to this variation. Since the rats’ diet in this study was standardised, it is unlikely to be the confounding factor. 

Tyrosine, categorised as a nonessential amino acid, is naturally synthesised within the body through the conversion of phenylalanine. In the current study, L-tyrosine is not detected in the OC rats, probably attributable to the heightened levels of phenylalanine in these rats, potentially impeding the formation of L-tyrosine. A study by Hugle et al. identified a distinctive L-tyrosine signature in the synovial fluid of OA patients using proton nuclear magnetic resonance [29]. Another study by Chen et al. reported elevated tyrosine levels in the serum of OA patients, suggesting that reducing L-tyrosine levels could have potential advantages in the management of OA [30]. It is worth noting that the supplementation of glucosamine sulphate appeared to downregulate the expression of L-tyrosine in the OG compared to OC, indicating that glucosamine might exert its effects by targeting L-tyrosine. However, there was an increase in the expression of L-tyrosine in the OGT when compared to OC, OG, and OT groups. This suggests that a combination of glucosamine sulphate and palm TRF may not be effective in reducing L-tyrosine levels in OA.

Arginine and proline metabolisms are the most reported amino acid metabolic pathways found to be associated with OA [30,31,32,33,34,35,36], and they are critical for collagen production. When cartilage is injured, the body initiates the healing process, and the arginine–proline metabolic pathway is activated to create proline for collagen production [37]. Metabolites involved in arginine and proline metabolism are L-arginine, L-proline, spermidine and creatine. Arginine is a semi-essential amino acid that is used to synthesise a variety of molecules, such as urea, nitric oxide, polyamines, proline, glutamate, creatinine, and agmatine [33]. Arginine could be considered a pivotal metabolite, as it serves as a precursor involved not only in arginine and proline metabolism but also in the nitric oxide production pathway, which has also been linked to OA [31]. Several studies have documented a decrease in arginine levels among individuals with OA [33,36,38]. A study conducted on a rabbit OA model induced by anterior cruciate ligament transection identified a correlation between elevated plasma arginine levels and lower histological severity scores for cartilage. This correlation implies that the diminished arginine concentration is likely a consequence of the OA and its progression [39]. The reduction in arginine levels among patients with OA possibly arose from heightened arginine breakdown [33], increased demand for arginine in cartilage repair in OA [40], and the body’s inability to meet the demand for arginine [36]. All of these conditions could result in limited proline availability for the collagen synthesis that is essential for cartilage repair, which in turn, could result in a decreased nutrient supply to the joint [22]. However, in the current study, serum analysis revealed an upregulation of L-arginine in the OC when compared to the SC, indicating an increase in L-arginine levels in rats with MIA-induced OA. According to McHugh [41] [], elevated L-arginine levels are found in inflammatory arthritis. Given that intra-articular MIA induction triggers transient inflammation [42], this might explain the heightened expression of L-arginine in rats with MIA-induced OA. However, it is worth noting that L-arginine expression was not observed across all treated groups. 

L-proline plays a vital role in collagen synthesis, which is essential for the health of tendons and joints [43]. Diminished arginine levels may limit proline availability [22]. Inversely, increased arginine levels due to inflammation could potentially elevate proline levels. In the present study, an increased proline level was found in the serum of OC compared to SC. On the other hand, the administration of glucosamine, TRF, and glucosamine + TRF downregulated proline levels in OA rats. Both glucosamine [8,44] and palm TRF [45] have been previously reported to possess anti-inflammatory properties. This suggests that their supplementation may have mitigated inflammation induced by MIA, subsequently reducing proline expression. 

Spermidine, a naturally occurring polyamine, has demonstrated potential benefits in inflammatory diseases when administered externally [46,47]. It functions as an antioxidant by scavenging free radicals and regulating other antioxidative mechanisms, thus protecting against oxidative stress [48,49,50]. However, it is important to note that spermidine, while acting as a free radical scavenger, can also produce harmful reactive oxygen species through its catabolism [51]. Previous research has shown that spermidine levels accumulate in the urine, synovial fluid, and synovial tissue of RA patients [52]. Similarly, increased spermidine levels have been detected in the serum of OA patients, which could indicate heightened oxidative stress or reduced activity of the enzyme spermine synthase responsible for converting spermidine into spermine [35], ultimately impairing lysosome function and leading to increased oxidative stress [50]. This observation aligns with the findings in the current study, where spermidine was found to be upregulated in OC rats compared to SC rats, which was possibly due to increased oxidative stress resulting from MIA-induced OA. A study by Silva et al. has revealed that the administration of exogenous polyamines under the hind paw surface can induce pain and swelling in non-arthritic rats, while inhibiting the synthesis of endogenous polyamines in arthritic rats can reduce inflammatory pain [53]. In our study, supplementation with glucosamine and TRF led to a reduction in spermidine levels in the OG and OT group compared to the OC group, indicating that these treatments may have inhibited the production of endogenous spermidine, thereby potentially alleviating inflammatory pain in the OA group. 

Creatine, a nitrogenous compound, is primarily synthesised in the liver with smaller amounts from the kidneys and pancreas. Comprising three amino acids—glycine, arginine, and methionine—creatine can also be obtained from our diets, notably from protein-rich sources like red meat and fish. It is stored in skeletal muscle in the form of creatine phosphate and plays a pivotal role in regenerating ATP from ADP following muscle contractions [54]. In a study by Neves et al. involving postmenopausal women with knee OA who engaged in strength training, creatine supplementation was found to enhance physical function, increase lower limb lean mass, and improve overall quality of life [55]. In the present study, creatine was not differentially expressed in the OC vs. SC. Interestingly, TRF supplementation was associated with a decrease in creatine expression. Specifically, reduced expression was observed in the OT in comparison to the OC, OG, and OGT, suggesting that TRF may offer benefits by lowering creatine levels via unknown mechanisms. 

Tryptophan metabolism, primarily occurring through the kynurenine pathway and mediated by intestinal immune and epithelial cells via the enzyme indoleamine 2,3-dioxygenase, plays a significant role in inflammation and neurotransmission [56]. Disruptions in tryptophan metabolism have been associated with OA and pain, which has opened avenues for the development of drugs targeting this pathway [56,57]. Typically, the targets for such treatments involve enzymes within these pathways, their metabolites, or their receptors [58]. Two metabolites affected in tryptophan metabolism are L-tryptophan and L-kynurenine. L-tryptophan, a plant-derived amino acid, is essential for in vivo protein synthesis and undergoes metabolic transformations into bioactive compounds like serotonin, melatonin, kynurenine, and the vitamin niacin (nicotinamide) following consumption [59]. Researchers such as Hu et al. have linked tryptophan and its metabolites, including kynurenic acid, to the ability to distinguish between OA and healthy control subjects based on synovial fluid analysis in rabbits [60]. Additionally, Huang et al. reported lower plasma L-tryptophan levels in patients with OA compared to healthy controls [61], suggesting that increasing L-tryptophan levels could be beneficial for patients with OA. In the current study, a downregulation of L-tryptophan and L-kynurenine levels in the OG compared to the OC was observed. However, L-tryptophan was upregulated in the OT and OGT when compared to the OC, while L-kynurenine was upregulated in the OT compared to the OC. These findings suggest that TRF may have a beneficial effect in elevating L-tryptophan and L-kynurenine levels in OA. 

Overall, OA induced by MIA had a discernible impact on the levels of various compounds within pathways related to the biosynthesis of phenylalanine, tyrosine, and tryptophan as well as the metabolism of phenylalanine, tryptophan, arginine, and proline (Figure 7). These pathways are classified within the amino acid metabolism category in the KEGG pathway map [62]. Interestingly, a previous study by Li et al. has established a close correlation between synovial hyperplasia, inflammation, and alterations in amino acid metabolism in OA rats [63]. Importantly, the administration of glucosamine and TRF effectively mitigated the influence of MIA-induced OA on these pathways, which was likely due to their anti-inflammatory properties.

## 4. Materials and Methods

### 4.1. Preparation of Treatment Solution

The palm TRF (EVNol™) used in this study was generously provided by ExcelVite Sdn. Bhd. (Chemor, Malaysia). It consisted of the following: 12% α-tocotrienol, 2% β-tocotrienol, 19.3% γ-tocotrienol, 5.5% δ-tocotrienol, and 11.9% α-tocopherol. Glycerol served as the excipient, and it was diluted with refined olive oil (BertolliTM, Deoleo, Cordoba, Spain) at a 1:10 ratio for oral administration. Refined olive oil, known to have minimal vitamin E content and other polyphenols [64], was selected because it does not influence the progression of OA, as evidenced in a study by Chin et al. [18]. Palm TRF was administered at a dosage of 100 mg/kg of body weight based on a previous study demonstrating its efficacy in preventing cartilage degradation in rats with OA [18]. Glucosamine sulphate powder (Rottapharma Ltd., Dublin, Ireland) was dissolved in distilled water, resulting in a solution with a concentration of 10 mg/mL, which was then orally administered at a dose of 250 mg/kg of body weight. This dosage has been shown to prevent cartilage damage and alleviate joint pain in rats with OA [65]. MIA was acquired from Sigma-Aldrich (St. Louis, MO, USA) and was dissolved in 50 µL of normal saline prior to intra-articular injection [18,66].

### 4.2. Animal Treatment

The animal-handling procedures were reviewed and approved by the Universiti Kebangsaan Malaysia Animal Ethics Committee (Approval code: FAR/PP/2018/KOK YONG/26-SEPT./946-JAN.-2019-DEC.-2020). The study adhered to the guidelines set forth by the Malaysian Animal Welfare Act (2015) for conducting animal experiments. For this study, male Sprague–Dawley rats, aged three months and weighing between 250 and 300 g, were procured from the laboratory animal resource unit of the authors’ institution. These rats were housed individually in plastic cages in a controlled environment with a temperature of 27 °C and a light–dark cycle of 14/10 h. They were provided with standard rat chow (702P, Gold Coin, Port Klang, Malaysia) and had unrestricted access to tap water. Following a period of seven days for acclimatization, the rats were randomly divided into five groups, each consisting of six rats. These groups included the sham control group (SC), OA control group (OC), OA group treated with palm TRF (OT), glucosamine sulphate (OG), or a combination of palm TRF and glucosamine sulphate (OGT). OA was induced in all groups, except for the SC, by performing intra-articular injections of MIA into the right knee joint of the rats using a 26 G needle. This procedure was carried out under anaesthesia with a combination of ketamine, xylazine, and Zoletil^TM^ cocktails. The oral supplementation regimen, administered twice daily, commenced the day following the MIA injection. The OG and OGT groups received glucosamine sulphate at a dosage of 250 mg/kg/day in the morning, while the remaining groups were administered an equivalent volume of normal saline. The OT and OGT groups received palm TRF at a dose of 100 mg/kg/day in the evening, while the other groups received an equivalent volume of refined olive oil. Following four weeks of treatment, the rats were humanely euthanised, and blood samples were collected through cardiac puncture while the rats were under anaesthesia with a cocktail of ketamine, xylazine, and Zoletil^TM^. The collected blood was centrifuged at 4 °C and 3000 rpm for 10 min to obtain the serum, which was subsequently stored at −70 °C until analysis.

### 4.3. Sample Preparation for Metabolite Extraction

For the extraction of metabolites, 100 µL of thawed serum was combined with 800 µL of 100% methanol in a glass tube. This mixture was briefly vortexed for 15 s, after which 1.6 mL of DCM was added. It was vortexed again for another 15 s before being subjected to centrifugation at room temperature, running at 300 rpm for 1 h. Subsequently, 600 µL of distilled water was introduced into the tube, which was followed by a 15 s vortexing and a 10-minute incubation at room temperature. The mixture was once more vortexed at 1000× *g* at room temperature for 10 min, causing it to separate into two distinct layers, namely the lower and upper phases. The upper layer (1 mL), which contained the metabolites, was collected, dried and stored at −80 °C.

### 4.4. Ultra-High Performance Liquid Chromatography–Tandem Mass Spectrometry (UHPLC-MS) 

The dried samples were reconstituted by adding 100 µL of a mixture consisting of isopropanol and methanol in a 1:1 ratio. After reconstitution, the samples underwent filtration using syringe filters (Titan3 Chromatography Syringe Filter, Thermo Scientific™, Waltham, MA, USA) and were subsequently transferred into glass vials. To create a quality control sample, 5 µL from each sample was collected in a tube, vortexed, and then 100 µL of the resulting mixture was transferred into a new glass vial. Blank samples were prepared by introducing 200 µL of distilled water into a separate glass vial. These samples, along with the QC and blank samples, were placed randomly into the UHPLC autosampler (UltiMate™ 3000, Thermo Scientific™, USA). The UHPLC system employed a C18 column (100 mm × 2.1 mm, 1.7 µm; Synchronis™, Thermo Scientific™, Waltham, MA, USA) as the stationary phase. The mobile phase A consisted of water with 0.1% formic acid, while the mobile phase B comprised acetonitrile (ACN) with 0.1% formic acid. Chromatographic separation was achieved at a flow rate of 450 µL per minute with the column temperature set at 55 °C. An injection volume of 2 µL was utilised. The elution gradients were programmed as follows: an initial 0.5% B for 1 min, followed by a gradient from 0.5% to 99.5% B over 15 min, holding at 99.5% B for 4 min, and concluding with a gradient from 99.5% to 0.5% B over 2 min. A series of blank samples were injected 15 times before the QC and serum samples were placed on the autosampler. The analysis was carried out using the UltiMate™ 3000 system with the Q-Exactive HF Orbitrap-MS (Thermo Fisher Scientific, USA) for untargeted metabolomics. The instrument operated under the following conditions: a sheath gas flow rate of 50 arbitrary units (AU), a gas flow rate of 18 AU, a sweep gas flow rate at 0 AU, S-lens set at 55 AU, capillary temperature at 320 °C, auxiliary gas heater temperature at 300 °C, and a spray voltage of 3.5 kV for positive mode and 3.0 kV for negative mode. MS scanning was performed at a resolution of 60,000 with a scan range spanning from 100 to 1000 (*m*/*z*). For MS/MS scans, a resolution of 15,000 was applied along with stepped normalised collision energy settings of 20, 40, and 60 AU. These settings were in accordance with those described by Goon et al. [67].

### 4.5. Statistical Analysis

Metabolomic profiles were subjected to comparative analysis using MetaboAnalyst 5.0 software (Xia lab @McGill, Montreal, QC, Canada). To conduct multivariate data analysis, principal component analysis (PCA) was employed. Subsequently, the datasets were scaled and subjected to processing to reduce potential technical variability among individual samples, following a methodology aimed at extracting relevant biological insights [68]. For univariate analysis, one-way ANOVA with Tukey’s HSD test as a post hoc analysis and fold-change analysis were carried out [69]. A statistical significance threshold of *p*-value < 0.05 was applied in this analysis. In pathway analysis, pathways were identified as significant when a *p*-value of <0.05 and an impact of >0.1 were reached. 4-hydroxy xylazine, ketamine/esketamine, dehydronorketamine, and xylazine were excluded from the pathway analysis because these metabolites were introduced during euthanasia.

## 5. Conclusions

The findings of these studies suggest that MIA-induced OA brings about changes in metabolic pathways related to arginine and proline metabolism, the biosynthesis of phenylalanine, tyrosine, and tryptophan, as well as the metabolism of phenylalanine and tryptophan. These pathways are fundamental aspects of amino acid metabolism. Notably, the utilisation of glucosamine and TRF supplements effectively alleviated the impacts of MIA-induced OA on these pathways, which was likely attributed to their anti-inflammatory properties. Hence, this research proposes that inflammation could be responsible for irregularities in the expression of intermediates within the amino acid metabolism pathway. Anti-inflammatory interventions capable of rectifying these abnormalities may offer potential for enhancing the management of OA. This research provides an understanding of the pathways that osteoarthritis (OA) may impact and the influence of TRF and glucosamine on these pathways. Yet, additional investigation is needed to determine whether these treatments directly or indirectly affect these pathways.

## Figures and Tables

**Figure 1 life-13-02343-f001:**
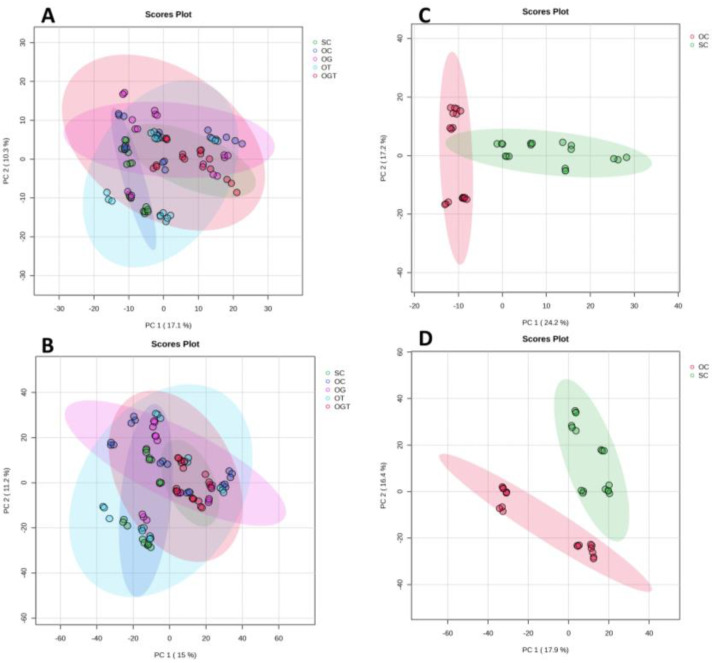
Distribution of samples for all groups and control groups based on PCA. PCA score plots between all groups in negative mode (**A**) and positive mode (**B**); and between OA control and sham control in negative mode (**C**) and positive mode (**D**). Abbreviation: SC, sham control; OC, OA control; OG, OA treated with glucosamine sulphate; OT, OA treated with palm TRF; OGT, OA treated with glucosamine sulphate and palm TRF.

**Figure 2 life-13-02343-f002:**
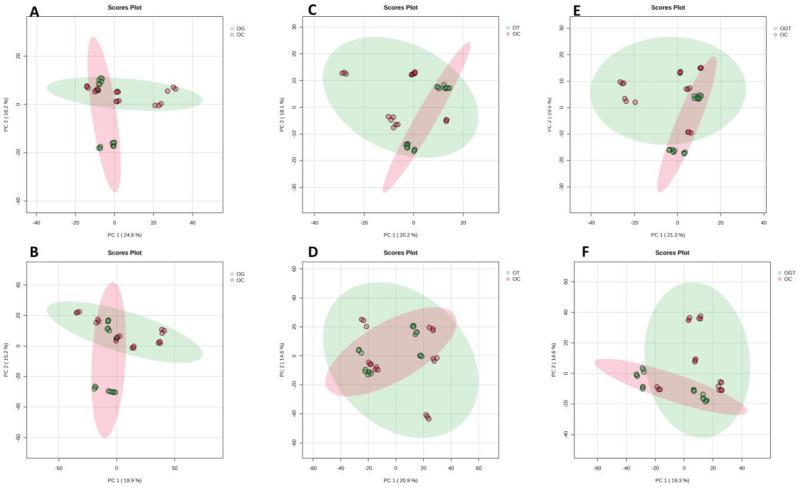
Distribution of samples for treatment groups versus vehicle-treated group based on PCA. PCA score plots between OG and OC in negative mode (**A**) and positive mode (**B**); OT and OC in negative mode (**C**) and positive mode (**D**); OGT and OC in negative mode (**E**) and positive mode (**F**). Abbreviation: OC, OA control; OG, OA treated with glucosamine sulphate; OT, OA treated with palm TRF; OGT, OA treated with glucosamine sulphate and palm TRF.

**Figure 3 life-13-02343-f003:**
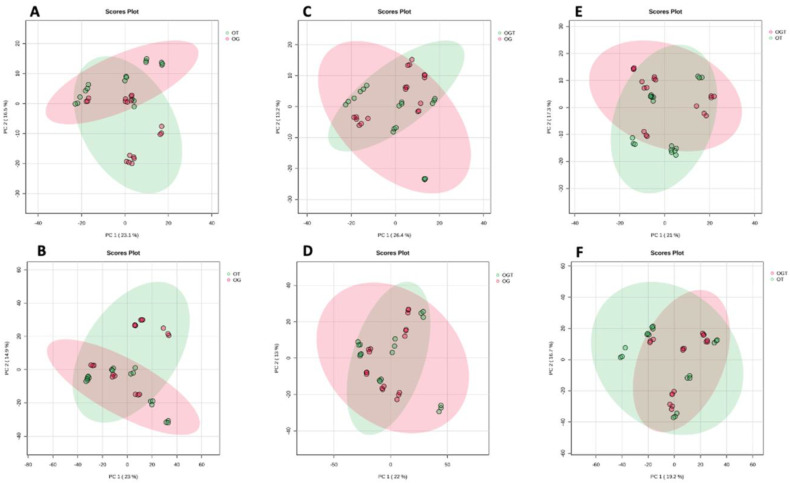
Distribution of samples for different treatment groups based on PCA. PCA score plots between OT and OG in negative mode (**A**) and positive mode (**B**); OGT and OG in negative mode (**C**) and positive mode (**D**); OGT and OT in negative mode (**E**) and positive mode (**F**). Abbreviation: OG, OA treated with glucosamine sulphate; OT, OA treated with palm TRF; OGT, OA treated with glucosamine sulphate and palm TRF.

**Figure 4 life-13-02343-f004:**
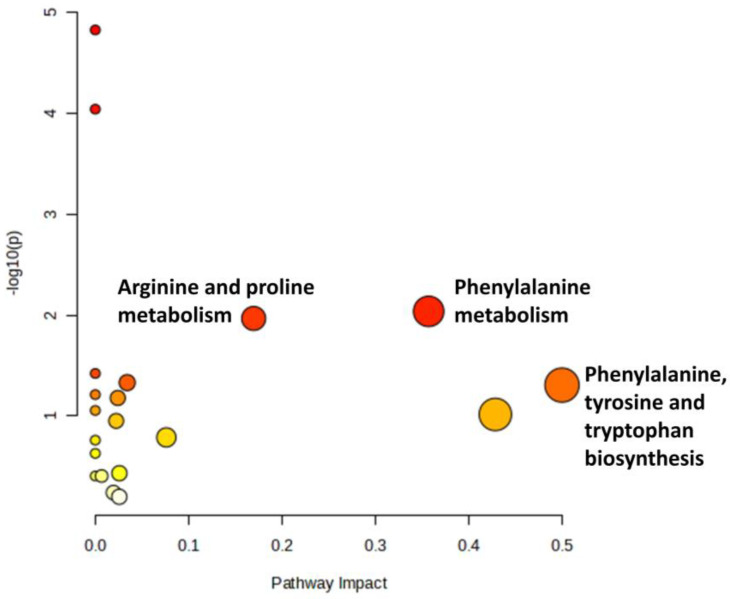
Biochemical pathway analysis of differentially expressed metabolites profiled in OA control vs. sham control. The degree of colour saturation (from white to red) indicates a rise in statistical significance, while the size of the circle changes based on the impact of the pathway.

**Figure 5 life-13-02343-f005:**
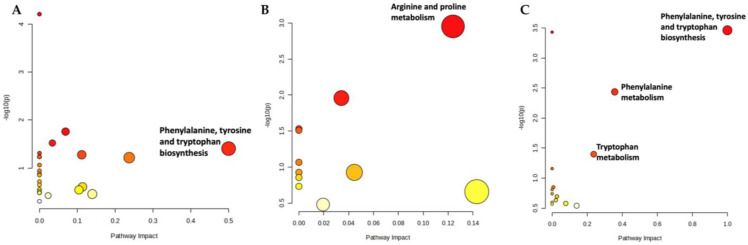
Biochemical pathway analysis of differentially expressed metabolites profiled in OG vs. OC (**A**), OT vs. OC (**B**) and OGT vs. OC (**C**). The degree of colour saturation (from white to red) indicates a rise in statistical significance, while the size of the circle changes based on the impact of the pathway. Abbreviation: OC, OA control; OG, OA treated with glucosamine sulphate; OT, OA treated with palm TRF; OGT, OA treated with glucosamine sulphate and palm TRF.

**Figure 6 life-13-02343-f006:**
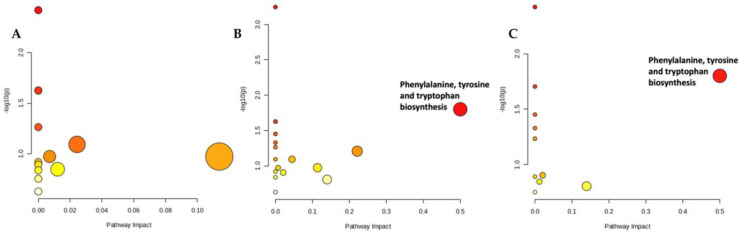
Biochemical pathway analysis of differentially expressed metabolites profiled in OT vs. OG (**A**), OGT vs. OG (**B**) and OGT vs. OT (**C**). The degree of colour saturation (from white to red) indicates a rise in statistical significance, while the size of the circles changes based on the impact of the pathway. Abbreviation: OG, OA treated with glucosamine sulphate; OT, OA treated with palm TRF; OGT, OA treated with glucosamine sulphate and palm TRF.

**Figure 7 life-13-02343-f007:**
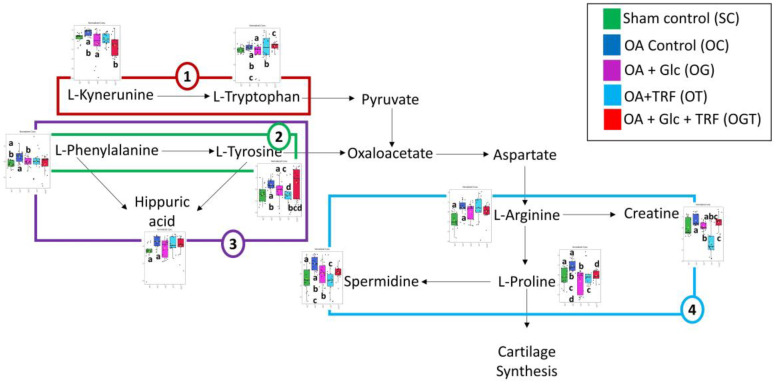
Pictograph showing alteration in tryptophan metabolism (1: red lines), phenylalanine, tyrosine and tryptophan biosynthesis (2: green lines), phenylalanine metabolism (3: purple lines) and arginine and proline metabolism (4: blue lines) in the serum of OA rats treated with glucosamine (Glc) and tocotrienol-rich fraction (TRF). Serum metabolite concentrations in the samples are depicted graphically and subjected to quantitative analysis. Box and whisker plots with 95% confidence intervals are presented for the quantified amino acids. Analysis was performed using one-way ANOVA with Tukey’s HSD post hoc analysis. Groups that share the same letter are significantly different compared to each other.

## Data Availability

Data are available on request from corresponding authors.

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
