# Peer review of "Serum Metabolomic Alteration in Rats with Osteoarthritis Treated with Palm Tocotrienol-Rich Fraction Alone or in Combination with Glucosamine Sulphate"

_life, 2023, doi:10.3390/life13122343_

Round 1

Reviewer 1 Report

Comments and Suggestions for Authors

The manuscript in question “Serum metabolomic alternation in rats with osteoarthritis treated with palm tocotrienol-rich fraction alone or in combination with glucosamine sulphate” has merit for publication.

It can benefit greatly from some edits. Major and minor suggestions are below.

 Major edits:

-Figure legends and results sections (ex. 2.1) should start with a premise as the title. Simply stating that differential (PCA) metabolic analysis was performed between the study groups is not sufficient. What do they signify? What is the meaning?

-Section 2.1 should start with a description of the study groups. The manuscript is written as if the reader has already read the methods.

-All of the front end is weak. Jumping straight into PCA without any preamble or context (with the exception of the abstract), is disorienting at best. It is purely descriptive and the meaning/significance of all the percentages is lost on the reader. No explanation of the significance of the differences is given, nor is it even clear, what the parameters are that are different. Is this PCA analysis of differences in metabolites? This must be stated. Some dialogue and narrative is required to ground this.

-Section 2.2 starts off talking about metabolites, shouldn’t this precede PCA results? Just to be clear, be sure to include upregulated in what group (line 120, 129, 135)? Down regulated in what group? OA or OC?

-Although results should exclude any discussion, it is generally good to include some overall assessment or minor conclusions at the end of each results section to sum up the overall lessons learned. This gives the reader directionality and the sense that something interesting was found. From your abstact it is clear that an effect of both treatments was observed, and this should be summed in the results, because you don’t want the reader to have to do an undo amount of detective work. The important points should be highlighted in the results and not buried in a table or a list of metabolites. Similarly with the pathway analysis.

-The discussion is quite thorough, however it would benefit from more focus on the results of this particular study. Many interesting studies are described in great detail. This is not appropriate for the discussion, and much of this can be removed. I would recommend compressing most of this background, most of which should be in the intro. Compress it down to general points that pertain to the issues that you need to discuss, and keep the references so that the reader can access more detail if desired. The discussion should be focussed mostly on a conversation and analysis of the results of this study, including strengths and weaknesses, with reference to pertinent literature, rather than a showcase for the literature. What is your study adding?

Line 24-26: The claim that TRF and glucosamine sulphate reverse the metabolic changes is justified. In the following line, you essentially repeat that this “suggests” that TRF and glucosamine reverse the metabolic changes. Instead, you could make some extrapolation about the mechanism or impact of these effects or leave it for the discussion.

Unless there is evidence that these treatments directly affect amino acid metabolism, it is important not to overstate the conclusion. The treatments might suppress OA by some unknown upstream mechanism, with metabolic fallout being a secondary effect. This should be the focus of the discussion. A better conclusion  (line 26-27) would be that reversal of OA induced metabolic effects are associated with the therapeutic action of TRF and glucosamine, (however it is not clear whether these are direct or indirect actions).

Minor edits:

Line 17: “rats, which...” or “rats that...”

Line 19: “or a combination of TRF and glucosamine sulphate (n=6)...”          

Line 48: “that has receive..”

Line 65: “..biomarkers for OA diagnosis, prognosis and therapeutics.”

Line 67: This line can be combined with the sentence on line 65, they make a similar point and same reference.

Line 73: Section 2.1 requires an introductory sentence. What data are the PCA scores comparing?

Line 77: OC, SC and other terms need to be defined in the results section, not just in the figure legend. The results section should be more than just describing the numbers, but should also state what these numbers signify, and which differences are statistically significant.

Line 173-175: exclusions should be moved to methods section.

Line239-240: this sentence is self evident. Yes, biosynthesis of amino acids is an important aspect of their metabolism. Maybe better to talk about amino acid precursors, and how their prevalence reflects on steady state levels of the amino acids in question. Is there anything special about these specific amino acids and with respect to their potential to affect Inflammation?

Line 263: could pain affect the eating behaviours of the mice?

Author Response

Thank you for your time to review this manuscript. We sincerely appreciate the feedback. We have carefully amended the manuscript accordingly. Please find our point-to-point response to the comments. We appreciate your favorable response to our action. Thank you!

Comments from Reviewer 1

The manuscript in question “Serum metabolomic alternation in rats with osteoarthritis treated with palm tocotrienol-rich fraction alone or in combination with glucosamine sulphate” has merit for publication.

It can benefit greatly from some edits. Major and minor suggestions are below.

  1. Major edits:
  2. Figure legends and results sections (ex. 2.1) should start with a premise as the title. Simply stating that differential (PCA) metabolic analysis was performed between the study groups is not sufficient. What do they signify? What is the meaning?

Response to reviewer

Thank you for your suggestion. We have updated based on your suggestion.

  1. Section 2.1 should start with a description of the study groups. The manuscript is written as if the reader has already read the methods.

Response to reviewer

Thank you for your suggestion. We have updated based on your suggestion (Lines 81-84).

  1. All of the front end is weak. Jumping straight into PCA without any preamble or context (with the exception of the abstract), is disorienting at best. It is purely descriptive and the meaning/significance of all the percentages is lost on the reader. No explanation of the significance of the differences is given, nor is it even clear, what the parameters are that are different. Is this PCA analysis of differences in metabolites? This must be stated. Some dialogue and narrative is required to ground this.

Response to reviewer

Thank you for your suggestion. This has been corrected. (Lines 86-94)

  1. Section 2.2 starts off talking about metabolites, shouldn’t this precede PCA results?

Response to reviewer

Thank you for your comment. The PCA serves as an initial visual assessment tool to detect changes in metabolite expression patterns before conducting a detailed analysis to identify upregulated or downregulated metabolites. Hence, PCA is presented as a preliminary step in the process.

  1. Just to be clear, be sure to include upregulated in what group (line 120, 129, 135)? Down regulated in what group? OA or OC?

Response to reviewer

Thank you for your suggestion. We have updated based on your suggestion (Lines 154, 155, 162, 164, 169, 171, 175 and 178)

  1. Although results should exclude any discussion, it is generally good to include some overall assessment or minor conclusions at the end of each results section to sum up the overall lessons learned. This gives the reader directionality and the sense that something interesting was found. From your abstact it is clear that an effect of both treatments was observed, and this should be summed in the results, because you don’t want the reader to have to do an undo amount of detective work. The important points should be highlighted in the results and not buried in a table or a list of metabolites. Similarly with the pathway analysis.

Response to reviewer

Thank you for your suggestion. We have updated based on your suggestion

  1. The discussion is quite thorough, however it would benefit from more focus on the results of this particular study. Many interesting studies are described in great detail. This is not appropriate for the discussion, and much of this can be removed. I would recommend compressing most of this background, most of which should be in the intro. Compress it down to general points that pertain to the issues that you need to discuss, and keep the references so that the reader can access more detail if desired. The discussion should be focussed mostly on a conversation and analysis of the results of this study, including strengths and weaknesses, with reference to pertinent literature, rather than a showcase for the literature. What is your study adding?

Response to reviewer

Thank you for your recommendation. We concentrated on the outcomes of the current study, substantiating them with references from prior literature. In the conclusion section, we included a statement underscoring the significance of the study. (Lines 522-525)

  1. Line 24-26: The claim that TRF and glucosamine sulphate reverse the metabolic changes is justified. In the following line, you essentially repeat that this “suggests” that TRF and glucosamine reverse the metabolic changes. Instead, you could make some extrapolation about the mechanism or impact of these effects or leave it for the discussion.

Response to reviewer

Thank you for your suggestion. These have been corrected (Lines 24-25)

  1. Unless there is evidence that these treatments directly affect amino acid metabolism, it is important not to overstate the conclusion. The treatments might suppress OA by some unknown upstream mechanism, with metabolic fallout being a secondary effect. This should be the focus of the discussion.

Response to reviewer

Thank you for your suggestion. The findings suggest that the treatments could be influencing amino acid metabolism. The nature of this effect, whether direct or indirect, remains uncertain, prompting the use of qualifiers like "could," "may," or "might." Nevertheless, we still have to discuss these results in the context of prior studies and attempt to draw connections based on the present study's findings.

  1. A better conclusion (line 26-27) would be that reversal of OA induced metabolic effects are associated with the therapeutic action of TRF and glucosamine, (however it is not clear whether these are direct or indirect actions).

Response to reviewer

Thank you for your suggestion. These have been corrected (Lines 25-27).

  1. Minor edits:
  2. Line 17: “rats, which...” or “rats that...”; Line 19: “or a combination of TRF and glucosamine sulphate (n=6)...” ; Line 48: “that has receive..”; Line 65: “..biomarkers for OA diagnosis, prognosis and therapeutics.”; Line 67: This line can be combined with the sentence on line 65, they make a similar point and same reference.

Response to reviewer

Thank you for your comment. These have been corrected

  1. Line 73: Section 2.1 requires an introductory sentence. What data are the PCA scores comparing?

Response to reviewer

Thank you for your comment. This has been updated as advised.

  1. Line 77: OC, SC and other terms need to be defined in the results section, not just in the figure legend. The results section should be more than just describing the numbers, but should also state what these numbers signify, and which differences are statistically significant.

Response to reviewer

Thank you for your comment. This has been corrected

  1. Line 173-175: exclusions should be moved to methods section.

Response to reviewer

Thank you for your comment. This has been moved to the method section (section 4.5)

  1. Line239-240: this sentence is self evident. Yes, biosynthesis of amino acids is an important aspect of their metabolism. Maybe better to talk about amino acid precursors, and how their prevalence reflects on steady state levels of the amino acids in question. Is there anything special about these specific amino acids and with respect to their potential to affect Inflammation?

Response to reviewer

Thank you for your comment. We opted to focus on the metabolites expressed in these pathways. Nevertheless, we did incorporate the impact of inflammation on the expressed metabolites associated with inflammatory processes.

  1. Line 263: could pain affect the eating behaviours of the mice?

Response to reviewer

Thank you for your comment. Certainly, the presence of knee pain can impact the eating behavior of mice or rats by restricting their ability to reach for food, leading to a reduction in food intake and, consequently, weight.

Reviewer 2 Report

Comments and Suggestions for Authors

This study presented several biomarkers by analyzing metabolites in OA using a non-targeted metabolic approach through blood analysis after administration of glucosamine and TRF in an OA model induced by MIA. In addition to previous studies, research results on phenylalanine, tryptophan, arginine, and proline metabolism and related pathways about OA were well presented. Below are the reviewer's comments.

Line76. Typo in value of PC1

Table 1: Please carefully check each value. Some values are not matched with each line.

Fig 7. The box plot data size is too small.  Please represent other ways.

Materials and Methods 4.4 Why were two voltages (3.5kV, 3.0kV) measured in UHPLC-MS, and why were the voltages positive and negative?

Comments on the Quality of English Language

English is fine.

Author Response

Thank you for your time to review this manuscript. We sincerely appreciate the feedback. We have carefully amended the manuscript accordingly. Please find our point-to-point response to the comments. We appreciate your favorable response to our action. Thank you!

Comments from Reviewer 2

This study presented several biomarkers by analyzing metabolites in OA using a non-targeted metabolic approach through blood analysis after administration of glucosamine and TRF in an OA model induced by MIA. In addition to previous studies, research results on phenylalanine, tryptophan, arginine, and proline metabolism and related pathways about OA were well presented.

Below are the reviewer's comments.

  1. Typo in value of PC1

Response to reviewer

Thank you for your comment. This has been corrected.

  1. Table 1: Please carefully check each value. Some values are not matched with each line.

Response to reviewer

Thank you for your comment. This has been corrected.

  1. Fig 7. The box plot data size is too small. Please represent other ways.

Response to reviewer

Thank you for your comment. We have tried to make the labellings more legible..

  1. Materials and Methods 4.4 Why were two voltages (3.5kV, 3.0kV) measured in UHPLC-MS, and why were the voltages positive and negative?

Response to reviewer

Thank you for your question. Mass spectrometers transform molecules into a charged and ionized state, and their detection relies on mass-to-charge ratios. The LC-MS/MS method includes two polarity modes: ESI MS negative mode and positive ion mode. At the capillary tip, an electrical field generates droplets of ionized compounds with either positive or negative charges. Consequently, two voltages, positive and negative, are employed to accommodate this dual charge state.